# Studies on the Effects of Fermentation on the Phenolic Profile and Biological Activity of Three Cultivars of Kale

**DOI:** 10.3390/molecules29081727

**Published:** 2024-04-11

**Authors:** Magdalena Michalak-Tomczyk, Anna Rymuszka, Wirginia Kukula-Koch, Dominik Szwajgier, Ewa Baranowska-Wójcik, Jacek Jachuła, Agnieszka Welman-Styk, Kinga Kędzierska

**Affiliations:** 1Department of Animal Physiology and Toxicology, Faculty of Medicine, The John Paul II Catholic University of Lublin, Konstantynów 1I Street, 20-708 Lublin, Poland; anna.rymuszka@kul.pl (A.R.); agnieszka.welman-styk@kul.pl (A.W.-S.); kinga.kedzierska@kul.pl (K.K.); 2Department of Pharmacognosy with Medicinal Plants Garden, Medical University of Lublin, 1 Chodźki Street, 20-093 Lublin, Poland; virginia.kukula@gmail.com; 3Department of Biotechnology, Microbiology and Human Nutrition, University of Life Sciences in Lublin, Skromna 8 Street, 20-704 Lublin, Poland; dominik.szwajgier@up.lublin.pl (D.S.); ewa.baranowska@up.lublin.pl (E.B.-W.); 4Department of Botany, Mycology and Ecology, Institute of Biological Sciences, Maria Curie-Skłodowska University, Akademicka 19 Street, 20-033 Lublin, Poland; jacek.jachula@mail.umcs.pl

**Keywords:** kale, lactic acid fermentation, HPLC-MS fingerprinting, antioxidant activity, cholinesterase inhibition, Caco-2 cell line, cytoprotective activity, anti-inflammatory activity

## Abstract

Fermentation is used not only to preserve food but also to enhance its beneficial effects on human health and achieve functional foods. This study aimed to investigate how different treatments (spontaneous fermentation or fermentation with the use of starter culture) affect phenolic content, antioxidant potential, and cholinesterase inhibitory activity in different kale cultivars: ‘Halbhoner Grüner Krauser’, ‘Scarlet’, and ‘Nero di Toscana’. Chosen samples were further tested for their protective potential against the Caco-2 cell line. HPLC-MS analysis revealed that the fermentation affected the composition of polyphenolic compounds, leading to an increase in the content of rutin, kaempferol, sinapinic, and protocatechuic acids. In general, kale cultivars demonstrated various antioxidant activities, and fermentation led to an increase in total phenolic content and antioxidant activity. Fermentation boosted anti-cholinesterase activity most profoundly in ‘Nero di Toscana’. Extracts of spontaneously fermented ‘Scarlet’ (SS) and ‘Nero di Toscana’ (NTS) showed cytoprotective properties, as revealed by the malondialdehyde (MDA), lactate dehydrogenase (LDH), superoxide dismutase (SOD), catalase (CAT), and glutathione (GSH) assays. Additionally, strong anti-inflammatory activity of NTS was shown by decreased release of cytokines IL-1β and TNF-α. Collectively, the conducted studies suggest fermented kale cultivars as a potential source for functional foods.

## 1. Introduction

Plant-based foods, including vegetables and their processed products, are rich sources of compounds known for their health-promoting effects on human health [1]. Among these foods, members of the *Brassica* genus, such as broccoli, head cabbage, cauliflower, and kale, hold a special place due to their year-round availability, ease of cultivation, affordability, and nutrient abundance. Notably, kale (*Brassica oleracea* var. *sabellica*), often dubbed a superfood, is a plant whose cultivation began in the Mediterranean about 2000 years ago [2]. The genetic and morphological diversity of kale has led to the existence of more than 120 varieties and cultivars, some with very local occurrences in different climatic zones [3]. Consequently, this diversity, combined with environmental factors and geographic location, contributes to the wide spectrum of phytochemicals, vitamins, minerals, and bioactive substances present in kale [4,5]. Among them, kale contains significant levels of phenolic acids (e.g., gallic, protocatechuic, caffeic, and gentisic acids), glucosinolates (e.g., glucoraphanin), and flavonoids (e.g., quercetin and kaempferol), all of which have been linked to various health-promoting properties [4,6]. The most extensively studied properties are antioxidant and free radical scavenging activities [7]. According to the literature, these attributes have been associated with multiple biological effects, including anti-inflammatory, neuroprotective, antibacterial, heart-protective, and anticarcinogenic effects [4,7,8].

Recently, the health-promoting aspects of plants have gained particular importance, along with the potential of using them in technological processes to create food products with unique characteristics. One such process is fermentation, which irreversibly alters the properties of a food product and is a method of food preservation. It impacts the physicochemical and microbiological properties as well as the overall content of bioactive compounds in food. Fermentation not only extends the shelf life of products but also enhances their nutritional value and digestibility, reduces toxicity, and improves their sensory attributes. Lactic acid bacteria (LAB), which are part of the indigenous microbiota of the raw food matrix, play a primary role in fermentation. LAB metabolize sugars into organic acids and alcohols through enzymatic transformations [9,10]. During fermentation, LAB can undergo either homofermentative or heterofermentative metabolism, which is strongly influenced by the type of microorganism. Research suggests that lactic acid fermentation is predominantly carried out by bacteria belonging to genera *Leuconostoc* (e.g., *L. mesenteroides*), lactobacilli (*Lactiplantibacillus plantarum*, *L. paraplantarum, Lacticaseibacillus rhamnosus*), *Lactococcus* (*L. lactis, L. lactis* subsp. *cremoris*), *Pediococcus* (*P. pentosaceus*, *P. acidilactici*), and *Weissella* (*W. cibaria*) [11,12]. A crucial factor in the effect of fermentation is whether it is spontaneous or achieved with the use of a starter culture. Spontaneous fermentation does not guarantee the consistent quality and targeted properties of fermentation achieved with a starter culture, but it serves as a source of microorganisms for developing new cultures for food fermentation. The interactions between microorganisms, ranging from amensalism to commensalism, competition, mutualism, and parasitism, have a profound impact on fermentation outcomes and product quality [13]. Therefore, increasing attention is being focused on the microbial species used in the starter culture to improve and control the quality of the final product. 

Scientific evidence unequivocally indicates that fermentation is a process that alters the polyphenol content of foods (e.g., an excellent review: [14]). However, numerous reports in the literature diverge in their findings regarding the direction (increase or decrease) of these changes. This variability can be attributed to the complexity of microbiota-mediated biotransformation of phenolic compounds, fermentation conditions, and the properties of the processed material. Plant phenolic compounds often exist in the form of glycosides, esters, or polymers, which limits their potential as efficient antioxidants. During fermentation, microorganisms can hydrolyze these complex compounds by producing enzymes like glucosidase, amylase, cellulase, tannase, chitinase, or lipase, thereby altering their bioavailability [15]. 

Another aspect of fermented foods’ biological activity that has gained attention in the last few years is their anti-cholinesterase activity and cytoprotective properties [16,17]. Some fermented vegetables have been shown to exhibit antiacetyl- and antibutyrylcholinesterase activities (which are regarded as neuroprotective) and the ability to decrease the release of pro-inflammatory cytokines. These effects are probably linked to their abundance of phenolic compounds and lactic acid-producing bacteria [14]. Nonetheless, the precise mechanisms underlying the alterations in these activities during fermentation remain unclear. 

To the best of our knowledge, no previous studies have concurrently examined the effects of lactic acid fermentation on the health-promoting attributes of different kale cultivars. Furthermore, despite extensive identification and characterization of intrinsic kale components, knowledge regarding metabolites synthesized during fermentation remains limited. 

Since numerous studies have demonstrated that lactic acid fermentation changes and mostly improves the phytochemical profile and biological activity of food, it seems reasonable to investigate this form of plant processing in terms of improving human health. 

Given the above, this paper aims to investigate the primary research hypothesis: the cultivar and method of fermentation have an impact on the phenolic profile and biological activity of kale. To test this hypothesis, we assessed the phenolic profile (using HPLC-ESI-QTOF-MS/MS), total phenolic content, and in vitro antioxidant activity, as well as anti- acetylcholinesterase (anti-AChE) and anti- butyrylcholinesterase (anti-BChE) activities, in kale samples derived from three cultivars: ‘Halbhoner Grüner Krauser’, ‘Scarlet’, and ‘Nero di Toscana’. Furthermore, selected samples were subjected to cytotoxicity, cytoprotective, and immunomodulatory tests. 

## 2. Results and Discussion

### 2.1. Phenolic Compounds Profile

Fermentation is one of the most conventional food processing methods utilized in the food industry. Thanks to numerous scientific studies, it is known that the role of this process extends beyond food preservation and brings about various positive health-related changes [18,19]. Vegetables, including kale, are a valuable source of phenolic compounds, and their processing through fermentation can modify the profile of these biologically active substances [15,20]. Kale is abundant in phenolic compounds, primarily flavonoids and phenolic acids [21]. Diversity among the cultivars translates into a distinct polyphenol profile in kale, as confirmed in previous studies [22]. The tentative identification that was performed based on a high-resolution mass measurement and the analysis of literature data revealed the presence of different groups of metabolites, including organic acids (i.e., quinic acid and citric acid), phenolic acids (i.e., chlorogenic, protocatechuic, and coumaroyl-quinic acid derivatives), tannins, glucosinolanes (i.e., glucobrassicin, desulfo-4-methoxyglucobrassicin, and desulfo-neoglucobrassicin), and flavonoids (i.e., quercetin, kaempferol, and naringin derivatives) (Table 1). The registered composition of kale samples was in accordance with some previously published data on the plant. Formerly, Liu et al. showed the presence of similar groups of metabolites in kale samples grown under windowsill and chamber conditions [23]. According to their studies, next to the highly represented group of flavonoids—the derivatives of quercetin and kaempferol, present in the forms of aglycons, glucosides, or diglucosides—the authors observed the occurrence of sinapinic acid and glucosinolates. In the herein tested samples, we also confirmed the presence of the latter metabolites, like glucobrassicin derivatives. Formerly, other glucosinolanes were also reported, like desulfo-glucoiberin or desulfo-sinigrin. Sulphur-containing compounds are well represented among the plant species that belong to Brassicaceae, which is also the case for kale [24,25]. Kale is a plant that synthesizes phenolic acids. The studies described herein underline the presence of several compounds from this group; however, the chlorogenic acid derivatives (i.e., chlorogenic, neochlorogenic, and (z)-chlorogenic acids) are present in the analyzed samples in the highest concentration. The identification of all compounds was conducted by comparing them to the current scientific literature, which confirmed the pronounced antioxidant capabilities of kale attributed to its diverse metabolite profile [26,27].

In this study, the developed chromatographic method provided clear mass spectra of the kale samples. The investigated extracts showed a large variety of metabolites. In the negative ionization mode, SB was found to differ the most from the remaining samples (Appendix A). Sinapinic acid was detected only in the following samples: HKS, HKC, NTS, and NTC (Table 1). Judging by the distribution of the identified metabolites among the tested samples, SB seems to be the least rich in the phenolic components. SC was the only one that contained naringin, whereas protocatechuic acid was absent in the non-fermented samples: HKB, SB, and NTB. 

While studying the qualitative composition of the analyzed extracts, the peak areas of every major component were written down and set together in Table 2. The data presented in the table can provide an overall impression of the differences in the content of the selected metabolites among the samples. They should be read horizontally, allowing for the comparison of the peak areas of a given compound with one another. In Table 2, the largest peak area was marked as 100%, and the remaining ones are expressed as percentages of the leading one. As different metabolites are characterized by various ionization capacities, direct comparison of the peak areas of different compounds with each other is subject to a large error. The obtained results show significant differences between the samples. The extracts HKB, SB, and NTB, which represent the fresh plant material, were found to be the least rich in the metabolites. The percentage contents of the determined secondary metabolites were much higher in the fermented samples. Unequivocally, the process of fermentation increased the bioavailability of polyphenols from the plant matrix. The complexity of the fermented plant product contributes to the intricacy of the processes and the changes occurring in this matrix. This is strongly associated with the presence and succession of microorganisms during fermentation, as well as the enzymes produced by them [19]. 

### 2.2. TPC and Antioxidant Activity

Numerous reports in the literature indicate that polyphenol content (TPC) increases as a result of fermentation of various foods by a wide range of microorganisms [28,29,30]. The observed trend can be explained by the production of metabolites and the increased release of phenolic compounds from the food matrix because of the breakdown of the original compounds through fermentation. Nonetheless, these parameters rely on the nature of the plant material and the unique capabilities of the particular starter culture employed. The total phenolic content (TPC) of the studied kale samples before and after fermentation is given in Figure 1. In general, the results indicate that the polyphenol content in the analyzed samples depends on the cultivar, whether the sample underwent processing, and the fermentation method. The highest TPC value was observed in the ‘Scarlet’ cultivar subjected to fermentation: 33.8 ± 0.4 and 39.7 ± 0.3 mg/g d.w. for SC and SS, respectively. The unprocessed kale samples from the three examined cultivars exhibited a lower TPC value compared to post-fermentation, both spontaneously and with the use of a starter culture. The highest increase in the parameter (TPC pre-fermentation vs. post-fermentation) under study was observed for HKC, followed by SS and NTS, with increases of 76%, 71%, and 68%, respectively. Spontaneous fermentation resulted in a greater increase in TPC compared to starter culture only for ‘Scarlet’, while for other cultivars, no statistically significant differences were observed between the fermentation methods. The influence of fermentation on TPC in kale has been observed previously [31,32]. However, the existing scientific reports have mainly focused on kale juice and compared the use of different bacterial strains for fermentation. Yet, there is a lack of comparisons that demonstrate changes during the fermentation of distinct kale cultivars. In our prior work, we successfully developed a starter culture for kale fermentation, the application of which resulted in a product with the best attributes [32]. In the mentioned study, fermentation led to an increase in polyphenol content compared to the raw matrix, with the highest value observed in spontaneously fermented products, at over four times higher. Similarly, Szutowska et al. [31] also observed that the use of a consortium of microorganisms for juice fermentation resulted in a higher increase in TPC than with the use of single LAB strains. The effect also depended on the strain used. Interestingly, for some products, a decrease in polyphenol content is observed during fermentation. For instance, this relationship is observed much more frequently in the case of cocoa beans than with other plants. Specifically, Albertini et al. [33] observed a significant reduction in polyphenol content, especially at the beginning of fermentation, considering that it may be caused by various factors, including the migration of soluble polyphenols into the liquid produced during fermentation, as well as enzymatic and non-enzymatic oxidation. In another study, researchers indicated that the polyphenol content depended on the bacterial strain initiating fermentation, resulting in different trends for the same raw material. For example, in the fermentation of avocado leaves with a strain belonging to the *L. mesenteroides* species, a 70% decrease in TPC was observed, whereas under the same conditions, the *L. plantarum* 9567 strain did not lead to any changes in polyphenol content expressed as TPC [34]. Taking into account the examples cited and our own observations, we can confidently say that different products of plant origin, as well as the conditions and method of fermentation will have a significant impact on the content of phenolic compounds in fermented products. 

Fermented *Brassica* vegetables, including kale, exhibit notable antioxidant properties due to the presence of naturally occurring phenolic compounds in fresh plants [35,36,37]. This is very often reflected in the results obtained from the determination of the phenolic content and profile of the extracts [37]. Tests for measuring antioxidant capacity exhibit distinct characteristics in their approaches to evaluating this activity. For instance, ABTS takes into account both lipophilic and hydrophilic compounds, while DPPH is particularly responsive to hydrophilic substances [38]. In the present study, these two distinct in vitro methods for assessing antioxidant activity were compared. As shown in Figure 1, the radical-scavenging capacity of DPPH and ABTS exhibited a comparable pattern during LAB fermentation, with a more pronounced elevation observed in the ABTS methods. Initial ABTS antioxidant activity (70.3 ± 2.0 for HKB; 75.0 ± 1.4% for SB, and 50.2 ± 2.6% for NTB) increased with fermentation. After spontaneous fermentation, this value most significantly changed for NTS, with an increase of over 24% compared to unprocessed material. In the case of using a starter culture, the highest 25% increase in ABTS antioxidant activity compared to the control was observed for the NTC sample. An increase of 18% was observed for the HKC sample, while the lowest increase of 12% was observed for the SC sample. In the study of Kim et al. [39], kale juice fermented with selected LAB strains resulted in a significant increase in antioxidant activity measured by the ABTS method, reaching its highest level in the presence of *Limosilactobacillus reuteri* and *L. fermentum* strains. However, in comparison to other representatives of the genus *Brassica*, studies of lactic acid fermentations of kale leaves are scarce. Our experimental results showed that the DPPH radical-scavenging activities of non-fermented samples, i.e., HKB, SB, and NTB, were 32.5 ± 0.9%, 49.4 ± 1.3%, and 22.2 ± 2.4%, respectively (Figure 1). The obtained data showed that lactic acid fermentation increased the DPPH radical-scavenging activity. In the case of ‘Scarlet’, the change was noticeable both for spontaneous and starter culture fermentation, while for ‘Halbhoner Grüner Krauser’ and ‘Nero di Toscana’, it occurred only when using a starter culture. The highest activity as regards the stable radical DPPH was found for the SS and SC samples. Additionally, as can be seen from the results, of the samples analyzed, the largest increase in DPPH activity against untreated raw material was observed for the NTC sample (increase in activity by 50%). 

In general, an increase in DPPH and ABTS activities was noted after fermentation. Additionally, among the tested samples, the highest antioxidant activity was noted for the ‘Scarlet’ cultivar both before and after the fermentation, while the highest increase in antioxidant activity after fermentation was observed for the ‘Nero di Toscana’ cultivar. Similarly to TPC, the differences in antioxidant activity (both ABTS and DPPH) are explained by the release of bioactive compounds from complex chemical structures as well as by the varying profiles of bioactive compounds in the plant material. The relationship observed in the study is reflected in the literature [14]. Besides, individual LAB bacteria involved in fermentation exhibit variable enzymatic activity with regard to phenolic compounds. The literature reports also indicate that fermentation can have a negative impact on the health-promoting properties of foods subjected to this process. For example, in the work of Montijo-Prieto et al. [34], the authors noted that the obtained plant-based product fermented with 10 different LAB strains after processing had lower antioxidant activity measured by the DPPH assay than the control. The reduction in antioxidant activity could be correlated with the oxidation and degradation of antioxidant compounds [40].

The lactic acid fermentation of vegetables, as suggested by certain studies, has the potential to enhance their ability to scavenge free radicals, which is closely correlated with the phenolic content and profile of the extracts [41]. According to the analysis performed, the TPC was related to the antioxidant activity (Table 3). Namely, a strong positive correlation was found between the TPC parameter and ABTS and DPPH assays. Similarly, the ABTS assay demonstrated a significant positive correlation with the DPPH test.

### 2.3. Cholinesterase Inhibitory Activity

Neurodegenerative disorders encompass a diverse spectrum of neurological conditions that profoundly affect the well-being of humans worldwide [42]. Given their escalating prevalence, there is an urgent imperative to explore innovative therapeutic interventions to ameliorate these disorders. One notably effective approach involves targeting specific enzymes implicated in the pathology, namely AChE and BChE, thereby mitigating associated symptoms. Plants rich in polyphenolic compounds have exhibited the ability to inhibit cholinoesterases [43,44,45]. Additionally, particular plant-based fermented foods, like soybean powder [46] and Sipjeondaebo-tang [47], decreased acetylcholinesterase activity in cell culture tests and/or animal experiments. However, scientific papers dealing with fermented *Brassica* vegetables are relatively limited. These studies indicate that extracts from plants of the *Brassica* genus exhibit variable cholinesterase inhibitory activity [26,48,49,50]. Interestingly, findings regarding the impact of the fermentation process on this activity are not clear, with some sources indicating a decrease [51] or increase [52] in the inhibitory potential.

Figure 2 depicts the effect of different kale cultivars and types of sample processing on the AChE and BChE inhibition values. All examined samples demonstrated the ability to inhibit the enzymes. To the best of our knowledge, this activity regarding fermented kale leaves has not been studied previously. The present results reveal that before fermentation, fresh leaves showed anti-AChE activity ranging from 7.7% to 10.3% and anti-BChE activity ranging from 5.4% to 11.4%. However, no statistically significant differences were calculated among the three cultivars before fermentation. Nevertheless, fermentation significantly improved the anti-AChE activity of ‘Scarlet’ and ‘Nero di Toscana’ cultivars, especially after spontaneous fermentation. In our study, the highest level of AChE inhibition (44.5 ± 2.3%) was observed for NTS. Similarly, Mollica et al. [49] noted a great ability to inhibit the AChE activity of the ‘Cavolo Nero’ cultivar (with elongated blue-green leaves similar to those of ‘Nero di Toscana’). In this study, a high inhibitory potential was noted for HKS, NTC, and SC (26.7 ± 2.7%, 24.4 ± 2.6%, and 20.2 ± 1.2%, respectively). These findings are in line with studies that showed a significant increase in inhibitory ChE activity during fermentation [47,52]. In contrast, the lowest value was found in SC, with a rate of 2.6 ± 0.6%. Based on the research results obtained, we can assume that fermentation may generate new metabolites against AChE in the extracts, but they depend greatly on the kale cultivar and bacterial strains associated with different fermentation types. This effect may be explained by the presence of compounds that have been described in the literature as strong AChE and BChE inhibitors [53,54]. Among them, rutin, kaempferol, protocatechuic, and sinapinic acid content significantly increased in tested samples after fermentation, as described earlier (see Section 2.1, Table 2). Some reports also indicate a strong association between TPC and the inhibition of AChE activity [44]. Unexpectedly, in this study, AChE or BChE activities showed moderate negative correlations to the TPC and antioxidant potential measured by ABTS and DPPH methods.

Regarding BChE activity, the inhibition was consistent with the results observed for AChE inhibitory activity. In this case, the inhibition rate was between 0.3% and 24.5% for SC and NTC, respectively. We found that statistically, there was no significant change in the butyrylcholinesterase activity of the fermented samples compared to the non-fermented control samples for the ‘Halbhoner Grüner Krauser’ and ‘Scarlet’ cultivars. Significantly greater inhibitory activity with respect to BChE was observed in the Nero di Toscana cultivar. Specifically, spontaneous fermentation resulted in a 3.4-fold greater suppression of BChE inhibition compared to the control (NTB). These results may suggest that during this type of fermentation, the biotransformation of certain compounds in the material into easily absorbable bioactive constituents occurs [55,56].

### 2.4. Cytotoxicity, Cytoprotective, and Immunomodulatory Effects of Extracts on Caco-2 Cell Line

Samples that demonstrated the best activity based on the previous studies (Section 2.2 and Section 2.3) were tested for their possible biological activity using the Caco-2 cell line. The SS sample was selected based on TPC and antioxidant activity, while the NTS sample was chosen for its high anti-cholinesterase activity.

The effect of different concentrations of extracts on the viability of the human colorectal adenocarcinoma Caco-2 cells is presented in Figure 3A. The results showed that both of the samples at concentrations up to 250 µg/mL showed no statistically significant differences in the toxicity of the tested cell line after 24 h of incubation. Extracts at a concentration of 500 µg/mL reduced the viability of the Caco-2 cancer cells. Therefore, only samples in the concentration range 50–250 µg/mL were used for further analyses. Many previous studies indicate relatively high anti-inflammatory activity among *Brassica* vegetables. These abilities encompass the capacity to inhibit the production of inflammatory cytokines and the function of pro-inflammatory enzymes [57,58], adjust the activities of antioxidant enzymes [59], and curb lipid peroxidation. However, there are considerably fewer available data regarding the anti-cancer properties of kale [5,49], especially concerning the different cultivars within the variety. Since ATP is the primary source of cellular energy in normal mammalian cells, a change in ATP levels is considered an indicator of mitochondrial oxidative damage [60]. It was found that both extracts, already at the lowest tested concentration of 50 µg/mL, induced a significant increase in cellular ATP levels, confirming the protective effect of kale extracts against H_2_O_2_ (Figure 3B). Furthermore, LDH (lactate dehydrogenase) is an enzyme found in normal cells and is released outside the cell when the cell is damaged or undergoing lysis. Hence, the LDH content in cell supernatant is one of the indicators for assessing membrane integrity and the degree of cell death [61]. The results show that LDH release in Caco-2 cells pre-treated with both extracts (50–250 µg/mL) decreased significantly compared with the control consisting of the cells after H_2_O_2_ exposure. In the case of ‘Scarlet’, the highest decrease in LDH release was observed for a concentration of 100 μg/mL (a decrease of 3.5 folds). For the ‘Nero di Toscana’, an extract at a concentration of 250 μg/mL caused a 3.2-fold decrease in LDH activity. In previous research, other species within the *Brassica* genus have been associated with elevated protective activity, characterized by reduced LDH release activity [41] from human cell lines.

An increasing number of studies indicate that elevated levels of reactive oxygen species (ROS) are emerging as a significant factor contributing to cellular damage in various chronic diseases [62]. Oxidative stress can, in turn, cause the generation of excessive amounts of ROS, which can result in the loss of mitochondrial function and cell apoptosis. Reactive oxygen species (ROS), including hydrogen peroxide (H_2_O_2_), superoxide anions, and hydroxyl radicals, have the potential to induce various alterations at the cellular level. These changes encompass, but are not limited to, lipid peroxidation, oxidation of DNA and proteins, disturbance of the body’s redox balance, and modification of normal cell function and morphology [63]. The antioxidant defense consists mostly of enzymes such as catalase (CAT), superoxide dismutase (SOD), and glutathione (GSH). Among them, CAT serves as a marker for peroxisomes, playing a role in the removal of superoxide radicals. Superoxide dismutase (SOD) is crucial for detoxifying superoxide radicals. Additionally, GSH is involved in eliminating excessive reactive oxygen species (ROS). In this study, the amount of these enzymes was reduced in H_2_O_2_—stimulated Caco-2 cells (Figure 4). In the cells previously incubated with SS and NTS extracts, we noted a lesser decrease or even slight increase in the amount of CAT and GSH. The CAT activity increased up to 47% compared to normal cells (Figure 4A). When the cells were treated with H_2_O_2_, the GSH activity decreased by approximately 60%, whereas when pre-treated with kale extracts, the loss in GSH activity in cells was limited even to 14% and 11% for SS and NTS, respectively, at a concentration of 250 µg/mL (Figure 4B). Additionally, SS and NTS extracts lowered the decrease in SOD activity at each of the tested concentrations except for 50 µg/mL for SS (Figure 4C). These protective effects are likely due to the antioxidant properties and phenolic profile of kale samples, as described previously in this paper. Similarly, in the study of Larocca et al. [57], the authors demonstrated that incorporating powdered cauliflower leaves into the diet enabled them to mitigate oxidative stress by lowering lipid-peroxidation levels and maintaining the activities of antioxidant enzymes such as CAT and SOD. Malondialdehyde (MDA), a product of lipid peroxidation, is referred to as a biomarker of oxidative stress, whose concentration increases significantly when cells are exposed to oxidative stimulation and, at high concentrations, leads to cell membrane damage. In line with this, as shown in Figure 4D, a significant increase by 72–83% in MDA accumulation was observed in Caco-2 cells treated with 250 µM H_2_O_2_ for 2 h, compared to the control. At the same time, this accumulation was significantly or even completely decreased by pre-treatment with SS and NTS extracts in the whole range of concentrations. 

Finally, we also evaluated the effects of selected extracts on the level of pro-inflammatory cytokines, i.e., IL-1β and TNF-α, which are potent inflammatory cytokines essential for the immune system’s response to inflammation. Based on the experimental work carried out, it can be concluded that, along with the increasing concentrations of NTS extract, the release of IL-1β by Caco-2 cells after LPS exposure was decreased in a dose-dependent manner. At the highest concentration of this factor (250 μg/mL), there was an approximately 15-fold decrease in IL-1β expression, while the level of released cytokines after pretreatment with SS extract, regardless of concentration, was the same. A similar relationship was observed for TNF-α, where inhibition of the release of this cytokine was noted to be approximately 10-fold in the presence of NTS at a concentration of 250 μg/mL, while for SS, it did not significantly change compared to controls. Similarly to our study, Sim et al. [64] demonstrated the pronounced suppressive activity of broccoli sprout extracts regarding TNF-α expression in RAW 264.7 macrophages, noting, in contrast to our study, the high inhibition of IL-1β. We hypothesized that this may be related to the variation in single polyphenol content and/or microbial composition of the two different fermented products. For example, we demonstrated that kale of the ‘Nero di Toscana’ cultivar contains the highest amount of kaempferol among tested samples, which previously has been associated with strong anti-inflammatory activity [65]. On the other hand, a reduction in cytokine level in the NTS sample may be associated with significantly higher inhibitory AChE and BChE activity in this sample (shown earlier in Figure 2), as previously described [66]. 

## 3. Materials and Methods

### 3.1. Plant Material and Bacterial Strains

For analysis, three kale cultivars: ‘Nero di Toscana’ (obtained from PlantiCo Company, Zielonki, Poland), ‘Scarlet’ (obtained from W. Legutko Company, Zielonki, Poland), and ‘Halbhoner Grüner Krauser’ (obtained from Torseed Company, Zielonki, Poland) were grown under the same conditions in southwestern Poland and harvested in early November 2022. Fresh leaves of each cultivar were washed with tap water and air-dried for 4 h at room temperature (18–20 °C). Subsequently, they were frozen at −80 °C until the analyses (raw material) or subjected to fermentation trials.

The starter culture, as described in our previous study [32], comprised: *Lactiplantibacillus* (*Lpb*.) *plantarum* 332 (KY883559), *Lpb. paraplantarum* G2114 (KY883562), and *Pediococcus pentosaceus* 2211 (KY883563). These strains were originally isolated from fermented curly kale leaves and thoroughly tested [67]. Identification of all strains was carried out using 16S rRNA gene amplification and recA gene multiplex analysis. Furthermore, their technological and functional attributes as starter cultures for fermentation have been extensively studied [32,67]. For further investigation, the bacteria were cultured in Man, Ragosa, and Sharp broth (MRS, Merck, Darmstadt, Germany) under anaerobic conditions at 30 °C for 48 h. 

### 3.2. Fermentation Trials

To investigate the impact of fermentation on the characteristics of kale cultivars, we subjected samples of kale to both (i) spontaneous fermentation and (ii) fermentation with a previously acquired starter culture [32]. For spontaneous fermentation, we initially prepared kale leaves by cutting them into small pieces, approximately 1 cm in length. These leaf fragments were then mixed with 2% salt (*w*/*w*) and placed in glass jars. On the other hand, the fermentation with the starter culture was conducted following the methods outlined in our earlier work [32]. In both cases, all samples were allowed to ferment for 14 days in separate jars at ambient temperature (approximately 18–20 °C). These fermentation conditions were selected based on our prior experiments with curly kale [27,32,67] and preliminary data assessing changes in the pH value of the resulting samples).

Immediately after fermentation, plant samples were stored at −80 °C overnight. After 24 h, each sample (non-fermented and after fermentation) was subjected to lyophilization and subsequently ground into a fine powder using a mortar and pestle. The freeze-dried plant material was then stored at −20 °C for further analyses. 

The kale samples prepared for further analysis included 3 cultivars, resulting in 3 different types of products, the description of which can be found in Table 4.

### 3.3. Extraction Procedure and Extracts Preparation

The extraction method was conducted by adapting the procedures described by De Montijo-Prieto et al. [34] with slight modifications. For this purpose, previously ground plant material was accurately weighed (0.1 g) and mixed with 5 mL of an 80% methanol/water solution *v*/*v*. Extraction was conducted in an ultrasonic bath (Emmi 20 HC, EMAG, Salach, Germany) at 40 °C for 15 min. Every 5 min, the samples were vortexed with a Vortex Mixer (Velp Scientifica, Usmate Velate, Italy) for 15 s. Next, the extracts were centrifuged for 10 min at 9000 rpm. Following this step, the extraction with the solvent was repeated two more times, each time collecting the supernatant. Then, the supernatants were combined and evaporated under reduced pressure. 

### 3.4. Analysis of Polyphenols by HPLC-ESI-QTOF-MS/MS

The fingerprinting of the analyzed extracts was performed using an Agilent Technologies high-performance liquid chromatograph with a mass detector. This HPLC of the 1200 Series (Agilent Technologies, Santa Clara, CA, USA) contained a degasser, a binary pump, an autosampler, a thermostat, and a mass spectrometer composed of a quadrupole, time-of-flight analyzers, and an electrospray ionizer. Every sample was injected six times—three times in the positive and another three times in the negative ionization modes. The operational parameters of the mass spectrometer were set as follows: a capillary voltage of 3000 V, fragmentor voltage of 110 V, skimmer voltage of 65 V, nozzle voltage of 1000 V, *m*/*z* range of 40–1000 Da, gas temperatures of 275 and 325 °C for gas and sheath gas, respectively, gas flows of 12 L/min, and nebulizer pressure of 35 psig. All kale extracts were filtered through a nylon syringe filter with a 0.1 µm diameter, injected onto a chromatographic column produced by Agilent Technologies (Zorbax Eclipse Plus RP-18, 150 mm × 2.1 mm, 3.5 µm pore diameter) (Agilent Technologies, Santa Clara, CA, USA) at a volume of 2 µL, and developed in a gradient of acetonitrile with 0.1% formic acid (solvent B) and 0.1% water solution of formic acid (solvent A). The gradient settings were as follows: 0–2 min: 1% B, 15 min: 20% of B, 25 min: 40% B, 30–35 min: 95% of B, 35.1–42 min: 1% B. The flow rate was set at 0.2 mL/min and the temperature of the thermostat was 20 °C. The Mass Hunter program (v. B.10.00) was used to record and handle the obtained data.

### 3.5. Determination of Total Phenolic Content and Antioxidant Activity

The total phenolic content and antioxidant activity were determined using common spectrophotometric assays with a multi-mode plate reader FLUOstar Omega (BMG Labtech, Ortenberg, Germany). 

To determine the total phenolic content (TPC) of the extracts, a reaction with the Folin–Ciocalteu (F–C) reagent (Sigma-Aldrich, St. Louis, MO, USA) was performed. Specifically, 10 μL of the extracts were mixed with 100 μL of freshly diluted F–C reagent (diluted with distilled water at a ratio of 1:10). After 5 min, the solution was further mixed with 100 μL of 7.5% Na_2_CO_3_ (Sigma-Aldrich) and incubated for 60 min before measuring the absorbance at 650 nm. Simultaneous runs were conducted with a water blank and gallic acid (GA, Sigma-Aldrich). The total phenolic content was then calculated as mg gallic acid equivalents (GAE) per 1 g d.w. of the plant material.

The scavenging of ABTS free radicals was conducted according to the procedure described by Miller et al. [68], with slight modifications. In this analysis, 100 μL of the extract was combined with 100 μL of distilled water and 900 μL of ABTS solution. This ABTS solution, with an absorbance of 0.7 ± 0.02, consisted of 7 mM ABTS (Sigma-Aldrich) and 2.45 mM potassium persulfate; it was prepared and allowed to stand for 24 h at room temperature. After that, 10 μL of each extract was allowed to react with 300 μL of the ABTS solution, and the A_sample_ readings at 734 nm were recorded at 30 s intervals until a plateau was reached. The reagent blank (A_blank_), substituting the sample with water, was employed. ABTS inhibition was calculated using the formula:% inhibition = 100 × (A_blank_ − A_sample_)/A_blank_(1)

The DPPH radical scavenging assay was conducted by combining 100 μL of appropriately diluted samples with 900 μL of a 0.06 mM DPPH solution. The decrease in absorbance (A_sample_) was measured after 30 min at 515 nm.

### 3.6. Determination of Cholinesterase Inhibitory Activity

The assessment of acetylcholinesterase (AChE) and butyrylcholinesterase (BChE) inhibition using a 96-well microplate reader was conducted following the procedure described by Baranowska-Wójcik et al. [69]. The test solution comprised the sample, ATCh (acetylthiocholine iodide), DTNB (5,5′-dithiobis-(2-nitrobenzoic acid)), and either AChE or BChE (all reagents except samples were in Tris-HCl buffer). Absorbance was measured at 405 nm after 30 min, with the negative control containing only a buffer. Each sample was tested in at least eight replicates, and all solutions were prepared in the same buffer. Results were presented as mean ± SD. 

### 3.7. Cell Cytotoxicity Assay

The Caco-2 cell line, derived from human colon adenocarcinoma, serves as a commonly employed in vitro representation of the small intestine. It was acquired from the European Collection of Authenticated Cell Cultures (Cat. No. 86010202). The cells were cultivated on DMEM medium (Sigma-Aldrich) enriched with 1% NEAA, 100 IU and 0.1 mg/mL P/S, and 10% FBS at 37 ℃, 5% CO_2_, and 95% relative humidity.

Different concentrations of tested extracts (SS and NTS) in the range of 50–250 µg/mL were used to determine their cytotoxicity on Caco-2 cells. A control was used without extract under the same conditions. The viability was determined by a neutral red (NR) uptake assay (Sigma-Aldrich) according to the manufacturer’s instructions. The amount of dye integrated into the cells, which is directly related to the total number of cells with the intact membrane, was determined using spectrophotometry at 540 nm [27].

### 3.8. Evaluation of the Protective Effect of the Extract

To assess the capacity of the extract to protect Caco-2 cells from ROS-mediated oxidative injury, cells were preincubated for 24 h in the presence of different concentrations of extracts. At the end of the preincubation time, the medium was changed before adding the oxidative stress-inducing agent, 250 µM H_2_O_2_, for 2 h. Different controls were used: (i) Caco-2 cells without any treatment (referred to as ‘normal cells’ in the diagram) and (ii) Caco-2 cells exposed to 250 µM H_2_O_2_ (referred to as ‘control’ in the diagram). 

#### 3.8.1. Detection of ATP Content and Cell Membrane Integrity

The CellTiter-Glo Luminescent Cell Viability Assay (Promega, Madison, WI, USA) was employed to measure total cell ATP, following the manufacturer’s guidelines. Initially, cells were seeded at a density of 1 × 10^6^ cells per well in 96-well plates and allowed to adhere overnight in a CO_2_ incubator at 37 °C. Once cells adhered, the medium was replaced either with fresh medium as a control or with medium containing the tested extracts. Subsequently, a lyophilized reagent containing luciferin and luciferase was added, and the plates were agitated on an orbital shaker for 2 min to induce cell lysis. The cells were then left to incubate for 10 min at room temperature to stabilize the luminescence signal, which was later quantified using a luminometer (FLUOstar Optima).

The CytoTox-ONE Homogeneous Membrane Integrity Assay Kit (Promega) was utilized to gauge the release of lactate dehydrogenase (LDH) from cells with compromised membranes, employing a fluorometric technique. The assay was carried out following the manufacturer’s instructions. Initially, cells were seeded at a density of 1 × 10^4^ cells per well and exposed to various concentrations of the tested extracts for 24 h. Subsequently, 50 µL of medium from the treated cells was combined with 50 µL of the CytoTox-ONE reagent. After a 10 min incubation at room temperature in the dark, the reaction was halted by adding 50 µL of the stop solution, and fluorescence was measured using a fluorometer (FLUOstar Optima) with excitation and emission wavelengths of 560 nm and 590 nm, respectively.

#### 3.8.2. Determination of Glutathione Levels GSH

The GSH-Glo™ Glutathione Assay was employed to detect and quantify reduced glutathione (GSH). Cells were plated at 1 × 10^5^ cells/mL in 96-well white microplates and allowed to attach overnight. Then, the medium was replaced with fresh culture medium containing the studied substances as described above and incubated for 24 h at 37 °C. After removing the media, the GSH-Glo reagent was added to the cells and incubated at room temperature for 30 min. Then, reconstituted luciferin detection reagent was added to each well and mixed briefly using the plate shaker, and luminescence was measured with the multi-mode microplate reader.

#### 3.8.3. Determination of Superoxide Dismutase (SOD) and Catalase (CAT) Activity

Caco-2 cells were placed in a 24-well plate (6 × 10^4^ cells/well). After incubation for 24 h, extracts at different concentrations were added to the cells, while H_2_O_2_ (60 μM) was added after 18 h and induced for 4 h. Following treatment, cells from wells were solubilized in a lysis buffer containing protease inhibitors for 30 min and sonicated on ice. The cell lysate was centrifuged at 14,000× *g* for 5 min at 4 °C, then supernatants were aspirated and stored at −80 °C until measurement of superoxide dismutase (SOD) and catalase (CAT) activity using the Superoxide Dismutase, SOD, Activity Assay Kit (Sigma-Aldrich), and Catalase Assay Kit (Sigma-Aldrich), according to the manufacturer’s protocol.

#### 3.8.4. Determination of Lipid Membrane Alteration

Lipid peroxidation was estimated based on the malondialdehyde (MDA) concentration, a final product of lipid peroxidation, using a commercially available Lipid Peroxidation (MDA) Assay Kit (Sigma-Aldrich), according to the manufacturer’s instructions. Caco-2 cells (at a density of 1.0 × 10^6^ cells/well) were treated with the studied extracts as determined previously. After 24 h incubation, cells were collected and homogenized on ice with MDA lysis buffer, and then homogenates were centrifuged at 650× *g* for 15 min, 4 °C. A thiobarbituric acid (TBA) solution was added to all samples and heated in boiling water for 15 min. After cooling, the mixture was centrifuged at 650× *g* for 10 min, the supernatant was separated, and then absorbance was measured at 535 nm. The values of MDA in the samples were calculated using MDA standards and expressed in nmol/mL.

### 3.9. Detection of Cytokine Production

Caco-2 cells were seeded in 24-well plates at a density of 0.5 × 10^6^ cells per well and incubated with various dilutions of extracts for 18 h. Following preincubation, lipopolysaccharide (LPS) was introduced as an inducer of proinflammatory cytokines (30 ng/mL, Sigma-Aldrich), and the cells were further incubated for 8 h. Subsequent to incubation, the culture medium was collected, centrifuged to remove cells and debris, and then stored at −80 °C until further analysis.

The levels of proinflammatory cytokines IL-1β and TNF-α in the supernatants obtained post-exposure were determined using the Human IL-1 beta ELISA Kit (ThermoFisher, Vienna, Austria) and the Human TNF-alpha ELISA Kit (ThermoFisher), following the manufacturer’s instructions. All standards and samples from cells treated with extracts, as well as cells stimulated with LPS alone (referred to as ‘control’ in the diagram), were analyzed in triplicate within the same batch. Cytokine levels in the tested samples were calculated using equations derived from standard curves prepared with standard solutions of IL-1β and TNF-α, respectively.

### 3.10. Statistical Analysis

Before conducting analyses, the data underwent a normality check using the Shapiro–Wilk test. The data that did not follow a normal distribution were subjected to arcsin (ABTS, TPC, AChE, and BChE) or square root (DPPH) transformation. A one-way ANOVA followed by Tukey’s HSD test at α = 0.05 was applied to determine the significance of differences between fermentation trials and cultivars. Spearman ranks correlation coefficients were calculated between the ABTS and DPPH antioxidant properties of the samples, TPC, AChE, and BChE. Data for assays performed on the cell line were analyzed using a one-way ANOVA with a Dunnett’s post hoc test. Statistical analyses were conducted using Statistica 13.3 software.

## 4. Conclusions

This study aimed to investigate the impact of fermentation on the composition and biological properties of kale samples. The obtained results confirm the beneficial properties of this processing method. According to the HPLC-MS fingerprinting, fermentation triggered better bioavailability of a range of phenolic components, probably due to an increased release of these substances from the plant matrix. Fermented samples were characterized by a more rich profile in comparison to the non-fermented ones. This was reflected in the higher TPC and antioxidant activity of these samples. ‘Scarlet’ and ‘Nero di Toscana’ fermented spontaneously (SS and NTS, respectively) exhibited cyto-protective properties; additionally, NTS demonstrated strong immunomodulatory activity as shown by the decreased release of cytokines IL-1β and TNF-α. Notwithstanding, the mechanisms behind the observed changes remain unknown and need further investigation. 

## Figures and Tables

**Figure 1 molecules-29-01727-f001:**
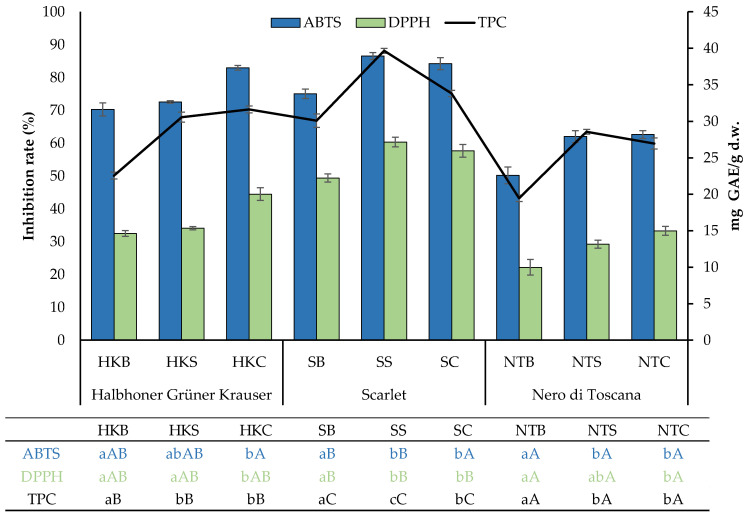
Effect of fermentation on TPC and antioxidant activity of kale samples. Bars represent the means ± SD of three replications. In the table, different uppercase letters indicate a significant difference for different kale cultivars (*p* < 0.05). Different lowercase and uppercase letters indicate significant differences between fermentation trials within a cultivar and between cultivars, respectively (Tukey HSD test at α = 0.05). ABTS—antioxidant activity against ABTS radical; DPPH—antioxidant activity against DPPH radical; TPC—total phenolic content.

**Figure 2 molecules-29-01727-f002:**
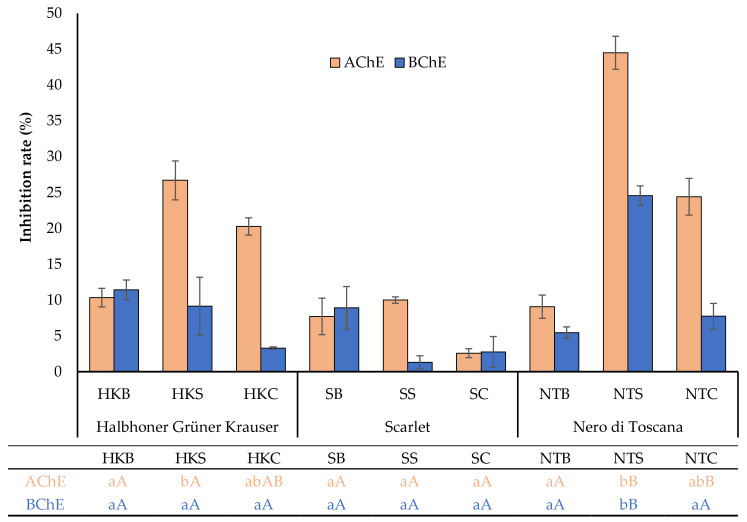
AChE and BChE inhibition rates (%) in the studied samples. Different lowercase and uppercase letters indicate significant differences between fermentation trials within a cultivar and between cultivars, respectively (Tukey HSD test at α = 0.05). AChE—acetylcholinesterase; BChE—butyrylcholinesterase.

**Figure 3 molecules-29-01727-f003:**
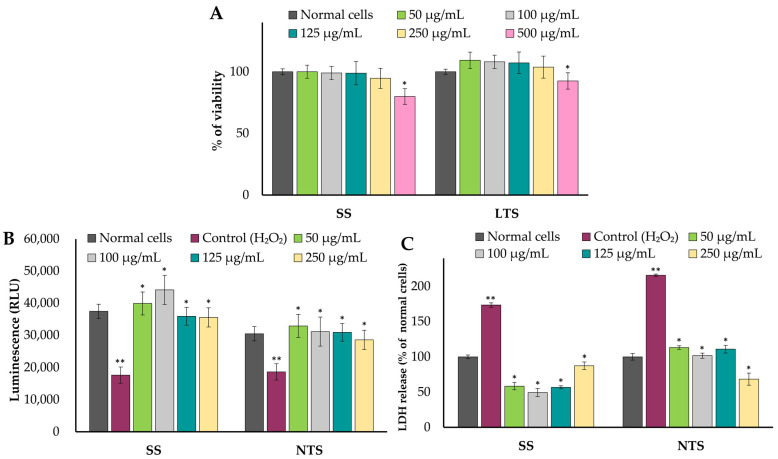
Effect of treatment with SS or NTS extracts on (**A**) viability of Caco-2 cells measured by neutral red uptake, ATP activity (**B**), and LDH release (**C**) after H_2_O_2_ exposure. Controls represent normal cells treated only with H_2_O_2_. Bars represent the means ± standard deviation (SD) of three replications; * indicates significant differences from the control (**) (H_2_O_2_-treated cells) calculated using one-way ANOVA followed by Dunnett’s test; SS—‘Scarlet’ cultivar after spontaneous fermentation; NTS—‘Nero di Toscana’ fermented spontaneously; RLU—relative light unit; LDH—lactate dehydrogenase.

**Figure 4 molecules-29-01727-f004:**
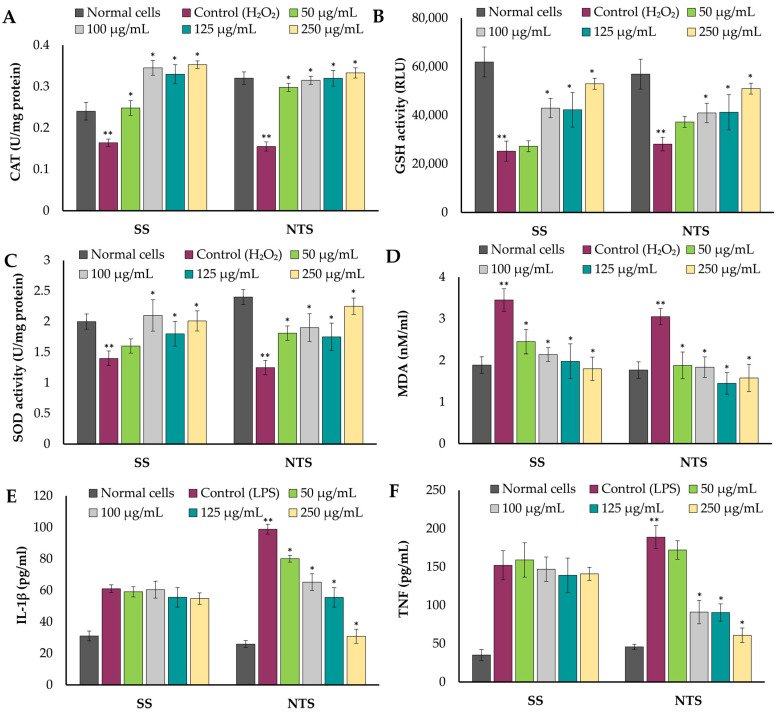
The levels of CAT (**A**), GSH (**B**), SOD (**C**), MDA (**D**) and pro-inflammatory cytokines IL-1β (**E**) and TNF-α (**F**) in Caco-2 cells treated with SS or NTS extracts in a range of concentrations from 50 to 250 and/or with control. In assays (**A**–**D**), H_2_O_2_-induced normal cells were used as a control group, while in assays (**E**,**F**), LPS-induced normal cells served as a control group. Each bar represents the mean ± standard deviation (SD); * indicates significant differences from the control (**) calculated using one-way ANOVA followed by Dunnett’s test; SS—‘Scarlet’ cultivar after spontaneous fermentation; NTS—‘Nero di Toscana’ fermented spontaneously.

**Table 1 molecules-29-01727-t001:** HPLC-ESI-QTOF-MS/MS fingerprinting of the analyzed extracts from kale (ion—ionization mode, Rt—retention time, delta—error of measurement, RDB—number of double bonds and rings in the structure).

No.	Ion. (+/−)	Rt. (min)	Molecular Formula	*m*/*z* Calculated	*m*/*z* Experimental	Delta (mmu)	RDB	Proposed Compound	Sample
1	−	2.19	C_7_H_12_O_6_	191.0561	191.0512	−0.46	3.5	Quinic acid	All
2	−	3.52	C_6_H_8_O_7_	191.0197	191.0207	−5.07	3	Citric acid	All
3	−	17.26/ 17.76/ 20.09	C_16_H_18_O_8_	337.0929	337.0927	0.57	8	Coumaroyl-quinic acid isomers	All
4	−	15.26/ 18.59	C_9_H_10_O_4_	181.0500	181.0505/ 181.0524/	0.73/−9.71	5	4-hydroxy-3-methoxy-phenylactetic acid (homovanillic acid)	All except SB
5	−	15.94	C_16_H_18_O_9_	353.0872	353.0877	0.3	8	Chlorogenic acid	All
6	−	16.61	C_16_H_18_O_9_	353.0872	353.0880	−0.55	8	Neochlorogenic acid	All
7	−	17.1	C_33_H_40_O_22_	787.1938	787.1944	−0.7	14	Quercetin-hexoside-dihexoside	All except NTC
8	−	17.59	C_7_H_6_O_4_	153.0202	153.0195	−1.09	5	Protocatechuic acid	All except HKB, SB and NTB
9	−	17.69	C_20_H_16_O_12_	447.0569	447.0539	6.69	13	Ellagic acid rhamnoside	All
10	−	18.86	C_16_H_18_O_9_	353.0872	353.0879	−0.27	8	(z)-Chlorogenic acid	All
11	−	19.2	C_16_H_20_N_2_O_6_S	367.0969	367.1005	−9.7	8	Desulfo-glucobrassicin	All except SC
12	−	21.8	C_27_H_30_O_17_	625.141	625.1425	−2.36	13	Quercetin dihexoside	All
13	−	23.11	C_27_H_30_O_16_	609.144	609.1482	−3.43	13	Rutin	All
14	−	24.28	C_21_H_20_O_12_	463.0882	463.0898	−3.45	12	Isoquercetin	All
15	−	24.44	C_17_H_22_N_2_O_7_S	397.1075	397.1035	10.04	8	Desulfo-neoglucobrassicin	All
16	−	24.85	C_11_H_12_O_5_	223.0626	223.0635	−10.28	5	Sinapinic acid	HKS, HKC, NTS and NTC
17	−	25.4	C_21_H_20_O_11_	447.0933	447.0946	−2.93	12	Quercitrin	All
18	−	28.113	C_17_H_22_N_2_O_7_S	397.1075	397.1039	9.03	8	Desulfo-4-methoxyglucobrassicin	All
19	−	29.69	C_15_H_10_O_7_	301.0348	301.0358	−1.4	11	Quercetin	All except SB
20	−	31.68	C_15_H_12_O_5_	271.0612	271.0639	−9.93	10	Naringin	SC
21	−	31.94	C_15_H_10_O_6_	285.0399	285.0404	0.22	11	Kaempferol	All except SB

**Table 2 molecules-29-01727-t002:** The average relative percentage content calculated based on the peak area value (*n* = 3) of the respective metabolites in the analyzed extracts of kale (The colours represent the percentage content value—green is the maximum measured peak area, whereas red pictures the minimum values).

Compound	Halbhoner Grüner Krauser			Scarlet			Nero di Toscana		
	HKB	HKS	HKC	SB	SS	SC	NTB	NTS	NTC
Quinic acid	8.05	20.86	19.90	100.00	7.75	8.78	10.06	12.65	25.13
Citric acid	51.60	20.21	46.55	100.00	26.27	10.89	37.78	14.38	8.46
Chlorogenic acid	100.00	96.63	9.96	25.12	48.55	53.00	59.89	36.51	42.55
Neochlorogenic acid	100.00	87.72	82.22	37.21	2.06	64.43	82.82	43.16	46.08
(z)-Chlorogenic acid	14.67	51.10	34.07	100.00	46.36	59.53	48.59	11.96	14.59
Coumaroyl-quinic acid isomer	27.77	45.21	29.89	15.07	13.16	13.48	43.75	72.97	100.00
Rutin	2.97	100.00	70.93	1.37	23.15	19.38	1.42	65.36	61.76
Isoquercetin	24.97	66.82	100.00	6.61	71.65	66.34	18.85	39.25	30.19
4-hydroxy-3-methoxy-phenylactetic acid	22.12	66.81	85.57	0.00	100.00	98.78	13.08	95.68	97.77
Protocatechuic acid	0.00	27.39	26.84	0.00	52.53	46.47	0.00	40.91	100.00
Quercetin	16.81	6.23	99.13	0.00	97.62	100.00	11.24	3.97	45.62
Kaempferol	5.41	70.21	53.84	0.00	29.08	31.86	8.25	100.00	76.40
Naringin	0.00	0.00	0.00	0.00	0.00	100.00	0.00	0.00	0.00
Sinapinic acid	0.00	100.00	96.41	0.00	0.00	0.00	0.00	63.91	71.51
Quercetin dihexoside	37.24	93.23	62.37	0.62	100.00	90.34	29.80	24.19	32.31
Quercetine-hexoside-dihexoside	64.09	11.67	10.83	100.00	26.71	17.16	33.95	5.36	0.11
Desulfo-neoglucobrassicin	29.85	31.93	54.19	21.34	100.00	81.79	22.19	42.32	56.05
Desulfo-4-methoxyglucobrassicin	16.11	100.00	96.41	11.85	65.42	61.32	12.52	82.66	98.87
Quercitrin	98.53	15.73	2.18	19.71	4.14	0.40	100.00	19.81	3.47
Desulfo-glucobrassicin	100.00	88.06	82.86	54.81	64.07	0.00	68.68	0.01	58.81

**Table 3 molecules-29-01727-t003:** Spearman ranks correlation coefficients between the antioxidant properties of the samples, their phenolic content, and their enzyme inhibitory activity.

	ABTS	DPPH	TPC	AChE	BChE
ABTS	1.00	0.93 **	0.91 **	−0.39 *	−0.61 **
DPPH	0.93 **	1.00	0.89 **	−0.44 *	−0.60 **
TPC	0.91 **	0.89 **	1.00	−0.20	−0.59 **
AChE	−0.39 *	−0.44 *	−0.20	1.00	0.49 **
BChE	−0.61 **	−0.60 **	−0.59 **	0.49 **	1.00

* significant at α = 0.05; ** significant at α = 0.01. ABTS—antioxidant activity against ABTS radical; DPPH—antioxidant activity against DPPH radical; TPC—total phenolic content; AChE—acetylcholinesterase; BChE—butyrylcholinesterase.

**Table 4 molecules-29-01727-t004:** The study’s methodology and kale sample codes.

Kale Cultivar	Type of Sample	Sample Code
‘Halbhoner Grüner Krauser’	Fresh leaves, before fermentation	HKB
	Leaves subjected to spontaneous fermentation	HKS
	Leaves fermented using a starter culture	HKC
‘Scarlet’	Fresh leaves, before fermentation	SB
	Leaves subjected to spontaneous fermentation	SS
	Leaves fermented using a starter culture	SC
‘Nero di Toscana’	Fresh leaves, before fermentation	NTB
	Leaves subjected to spontaneous fermentation	NTS
	Leaves fermented using a starter culture	NTC

## Data Availability

Data is contained within the article or Appendix A.

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
