# Peer review of "Studies on the Effects of Fermentation on the Phenolic Profile and Biological Activity of Three Cultivars of Kale"

_molecules, 2024, doi:10.3390/molecules29081727_

Round 1

Reviewer 1 Report

Comments and Suggestions for Authors

The manuscript presents an insightful investigation into the effects of fermentation on phenolic content, antioxidant potential, and cholinesterase inhibitory activity in different kale cultivars. Fermentation is a traditional method utilized not only for food preservation but also to enhance health benefits, particularly in the development of functional foods. The study examines the impact of spontaneous fermentation versus fermentation with a starter culture on various parameters.

Author Response

First of all, please check your spelling and grammar throughout the manuscript. There are some typos and errors that need to be corrected. You can use a spellchecker or a proofreading tool to help you with this task.

Appropriate corrections throughout the manuscript have been made. 

Secondly, your abstract is too long and does not clearly state the main findings and implications of your study. You should aim for a concise and informative summary of your research question, methods, results, and conclusions.

Abstract has been shortened and reorganized according to the reviewer's suggestion (Lines: 17-32).

Thirdly, your introduction should provide more background information and context for your study. You should explain why fermentation is important for food preservation and functionality, and what are the main mechanisms and factors involved. You should also review the relevant literature on the effects of fermentation on phenolic compounds, antioxidant activity, and cholinesterase inhibitory activity in different plant foods, especially kale. You should also define the key concepts and terms that you use in your study, such as phenolic compounds, antioxidant activity, cholinesterase inhibitory activity, Caco-2 cells, etc. You should also state your research objectives and hypotheses clearly and concisely at the end of the introduction.

The paper covers multiple aspects, including fermentation as a potential method to obtain functional foods, and the impact of fermentation on selected parameters of its biological activity. We tried to introduce each of them, but indeed some aspects have been overlooked. We have added the missing part about anti-cholinesterase and cytoprotective activities (Lines: 89-96). The hypothesis and the research objectives are in the last paragraph in the Introduction (Lines: 106-113).

Fourthly, your materials and methods section should provide more details and clarity on how you performed your experiments. You should describe the source, preparation, and storage of the kale samples, the fermentation conditions and parameters, the analytical methods and instruments, the cell culture and treatment protocols, and the statistical analysis. You should also report the units, ranges, and standard deviations of your measurements. You should also cite the recent references for the methods that you used or adapted from previous studies.

We appreciate your suggestions for enhancing the materials and methods section. However, we believe that our current description adequately covers the methodology employed in our study. While we understand the importance of providing detailed information on experimental procedures, it's important to note that our paper is extensive and quite long, and including overly detailed descriptions in the manuscript may not be necessary. Whenever the methods were of commonly known standard, we only briefly described them, but added relevant citations for more details. We have also already provided relevant citations for the methods used or adapted from previous studies to ensure transparency and accuracy.

Some of the information you mentioned as lacking is indeed described in detail in our manuscript (the source, preparation, and storage of the kale samples - lines: 451-457; fermentation conditions – lines: 470-478). However, we agree that the description of instruments used, cell line trials and statistical analyses were not fully explained. We have carefully reviewed your comments on this matter and have ensured that essential information has been appropriately included in the manuscript (Lines: 490-491, 557-559, 638-640). 

In the result part Figures 3 and 4 are not clear try to improve its quality.

Indeed, these Figures in the pdf file seem to be of poor quality. However, the original figures are in line with the journal requirements and were additionally uploaded in the system as separate files in high quality. 

Reviewer 2 Report

Comments and Suggestions for Authors

Dear Authors,
In my opinion, the article submitted for review entitled "Studies on the effects of fermentation on the phenolic profile and biological activity of three cultivars of faeces" is well organised and addresses currently attractive topics.
The manuscript presents a study on different curly kale cultivars on which the effects of the fermentation process were tested and selected parameters were used as research markers:  antioxidant potential, phenolic content,  and cholinesterase inhibitory activity. The study compares the effects of spontaneous fermentation and fermentation with starter culture.
Abstract - please follow the journal requirements - short description, aim, methodology and description of main results/conclusions.

The literature review focuses more on the information related to fermentation and starter cultures, and I missed a description of the later discussed issues such as the cytoprotective properties of fermented vegetables (here kale).
The studies in the methodology are clearly described and the necessary data are included in the individual methods.

The studies described in the methodology are written clearly, and the necessary data are included in the individual methods.
Regarding the results and their discussion, the tables and graphs display the collected information clearly, and their presentation is acceptable. The only issue relating to Figure 1 is that the TPC line may be misleading, as it shows the potential relationships that could arise between the culture of kale and/or the bacteria used. I would ask you to replace the showing from this parameter with a point-based one (as long as this arrangement does not obscure the showing of the results).

Author Response

Abstract - please follow the journal requirements - short description, aim, methodology and description of main results/conclusions.

Thank you for pointing this out. Abstract has been shortened and reorganized according to the reviewer's suggestion (Lines: 17-32).

The literature review focuses more on the information related to fermentation and starter cultures, and I missed a description of the later discussed issues such as the cytoprotective properties of fermented vegetables (here kale).

Relevant information in the Introduction section has been added according to the Reviewer's suggestion (Lines: 89-96). 

Regarding the results and their discussion, the tables and graphs display the collected information clearly, and their presentation is acceptable. The only issue relating to Figure 1 is that the TPC line may be misleading, as it shows the potential relationships that could arise between the culture of kale and/or the bacteria used. I would ask you to replace the showing from this parameter with a point-based one (as long as this arrangement does not obscure the showing of the results).

While the linear presentation of the TPC parameter might be confusing at first glance, for this case it was the clearest way to show the changes between different types in fermentation for each kale cultivar. We tried the point-based graph, but the arrangement obscured the showing of the results.  

Round 2

Reviewer 1 Report

Comments and Suggestions for Authors

The manuscript is suitable for publication in the current form.